# The effects of protein supplementation, fumagillin treatment, and colony management on the productivity and long-term survival of honey bee (*Apis mellifera*) colonies

Michael Peirson[1]*, Abdullah Ibrahim[1], Lynae P. Ovinge[2], Shelley E. Hoover[2¤a], M. Marta Guarna[1], Andony Melathopoulos[3¤b], Stephen F. Pernal[1]*

1 Agriculture and Agri-Food Canada, Beaverlodge Research Farm, Beaverlodge, Alberta, Canada, 2 Alberta Agriculture and Forestry, Lethbridge Research Centre, Lethbridge, Alberta, Canada, 3 School for Resource and Environmental Studies, Dalhousie University, Halifax, Nova Scotia, Canada

¤a Current address: Department of Biological Sciences, University of Lethbridge, Lethbridge, Alberta, Canada
¤b Current address: Department of Horticulture, Oregon State University, Corvallis, OR, United States of America
* steve.pernal@agr.gc.ca (SFP); mdjpeirson@gmail.com (MP)

## Abstract

In this study, we intensively measured the longitudinal productivity and survival of 362 commercially managed honey bee colonies in Canada, over a two-year period. A full factorial experimental design was used, whereby two treatments were repeated across apiaries situated in three distinct geographic regions: Northern Alberta, Southern Alberta and Prince Edward Island, each having unique bee management strategies. In the protein supplemented treatment, colonies were continuously provided a commercial protein supplement containing 25% w/w pollen, in addition to any feed normally provided by beekeepers in that region. In the fumagillin treatment, colonies were treated with the label dose of Fumagilin-B® each year during the fall. Neither treatment provided consistent benefits across all sites and dates. Fumagillin was associated with a large increase in honey production only at the Northern Alberta site, while protein supplementation produced an early season increase in brood production only at the Southern Alberta site. The protein supplement provided no long-lasting benefit at any site and was also associated with an increased risk of death and decreased colony size later in the study. Differences in colony survival and productivity among regions, and among colonies within beekeeping operations, were far larger than the effects of either treatment, suggesting that returns from extra feed supplements and fumagillin were highly contextually dependent. We conclude that use of fumagillin is safe and sometimes beneficial, but that beekeepers should only consider excess protein supplementation when natural forage is limiting.

**Data Availability Statement:** All relevant data are within the paper and its Supporting Information files.

**Funding:** Agriculture and Agrifood Canada, https://www.agriculture.canada.ca, supported this project through Project # J-000049 "Health of Bee Pollinators in Canadian Agriculture", which was awarded to Steve Pernal (SFP). The funders had no role in study design, data collection and analysis, decision to publish, or preparation of the manuscript.

**Competing interests:** The authors have declared that no competing interests exist.

## Introduction

In recent years, high rates of honey bee *(Apis mellifera* L.) colony death, queen loss, and poor colony growth have often been reported [1–3] and many have sought explanations [4]. Surveys of beekeepers often report that environmental and management factors such as poor queens, starvation, and small colony size are leading causes of death [1, 3]. Others have summarized major factors in terms of the four 'P's—poor nutrition, pesticides, pathogens, and parasites [5, 6]. One parasite in particular, *Varroa destructor* Anderson & Trueman, is recognized as extremely important [7] and a variety of controls have been developed against it [8]. Some stressors, which include acute pesticide exposures, sustained pollen deprivation, and certain pathogens, produce identifiable symptoms. Others, such as covert viral infections [9] or nosema disease [10], can be largely invisible. As beekeepers cannot respond to problems they cannot identify or predict, many turn to preventative or prophylactic solutions, often having uncertain efficacy.

Protein supplements and the anti-fungal medication fumagillin are two of the most well-established products used by beekeepers in Canada to promote colony health. Typically, protein supplements are supplied to colonies in spring before the first natural pollen is available [11], though they may also be fed in fall [12], while fumagillin is routinely supplied in sugar syrup prior to winter [11].

Protein supplements are intended to compensate for a shortage or nutritional inadequacy in the pollen available to hives. Supplementation is a longstanding practice recommended to beekeepers when natural pollen may be limited [13]. Commercial protein supplements such as the one used in this study are often based on soy flour, brewer's yeast, and sugar syrup, as recommended by Haydak [14–16], with a proportion of bee-collected pollen, although numerous other ingredients may be employed. Protein supplements for bees are popular and have been widely investigated in controlled conditions and short-term field studies, with inconsistent results [17–20].

Fumagillin is a treatment for the parasites *Vairimorpha apis*, and *Vairimorpha ceranae* (previously known as known as *Nosema apis* and *Nosema ceranae*) [21], which can shorten the lifespan of bees [17, 22–24]. *V. ceranae*, which is widespread in Canada [25], has been variously described as a severe [22] or a trivial [26] threat to honey bee colonies. Fumagillin has long been recommended for widespread prophylactic use [27], but some have questioned its effectiveness in controlling *V. ceranae* [28, 29]. In field trials, some authors have failed to detect colony-level economic benefits [30], but others have found increases in colony survival, size, and honey production [31]. Nearly all have demonstrated reductions in *Vairimorpha* prevalence or spore load [31, 32].

Between 2014 and 2016, we conducted a cohort study in which we attempted to evaluate the relative importance of nutrition, pesticides, parasites, pathogens, and general management in three Canadian beekeeping operations representing different regions and commercial beekeeping practices. Two treatments were applied in a factorial design: (1) protein supplements applied (or not applied) continuously throughout most of the active beekeeping season, as a test of the hypothesis that nutritional deficiency was a cause of colony under-performance, and (2) a fall application of fumagillin applied (or not applied) to control nosema disease. In this, our first output from this large-scale investigation, we discuss the effects of the treatments on colony productivity and survival and examine differences across beekeeping regions. Our expectation was that both treatments would improve colony health and, when in used in combination, would have additive beneficial effects across all locations.

## Methods

The trial was a fully crossed design with three locations and two treatments: 1) continuous protein supplementation, and 2) fall treatment for nosema disease. The locations were sites in

Canada which represented three common economic models of commercial beekeeping: hybrid canola seed pollination (*Brassica napus* L.) with limited honey production in Southern Alberta (SAB; 117 colonies), honey production from fields of commodity canola in Northern Alberta (NAB; 123 colonies), and lowbush blueberry (*Vaccinium angustifolium* Aiton) pollination with no surplus honey production in Prince Edward Island (PEI; initially 76 colonies, increasing to 122 after colony splitting). The colonies in Northern Alberta belonged to Agriculture and Agri-Food Canada (AAFC) and were managed by personnel in the apiculture research program, while colonies at the other locations were owned and managed by commercial beekeepers.

See S1 File and S1 Table for additional details.

## Protein supplement treatment

Beekeepers applied their normal early spring and fall feeding practices, which for Alberta producers but not PEI included providing commercial protein supplements (Global Patties with 15% pollen; Global Patties, Airdrie, AB, Canada) to all colonies in spring. Subsequently, half the colonies at each apiary received no additional supplement, while the other half received protein patties prepared by Global Patties according to their standard recipe, but modified to include 25% pollen from Canadian prairie sources. The modified (w/w) recipe contained: sucrose syrup: 46%, distillers dried yeast: 15%, defatted soy flour: 14%, irradiated pollen: 25%. The supplement was provided continuously as consumed throughout the active beekeeping season, except during the canola bloom, when pollen was known to be abundant (S2 and S3 Tables).

Pollen used in the 25% protein supplement was trapped corbicular pollen from the Canadian prairies, and was irradiated (Iotron Technologies Corp, Port Coquitlam, BC, Canada) at 10 kGy before use. Major constituents of the pollen included *Brassica napus* L, *Trifolium* spp., *Melilotus* spp., and *Glycine max* (L) Merrill.

## Fumagillin treatment

Each fall, Fumagilin-B® (DIN: 02231180; Medivet Pharmaceuticals, High River, AB, Canada) was applied to half the colonies per apiary (S2 Table). The method and timing of application varied among commercial beekeepers (S4 Table) but was always administered at the fall label dose (200 mg fumagillin per colony). After treatment with fumagillin-treated syrup, or throughout the fall if fumagillin was delivered as a syrup drench, unmedicated 67% (w/w) sucrose syrup was supplied *ad libitum* until the colonies were wrapped for winter or the hives stopped accepting additional feed. Syrup was supplied to each colony individually in hive-top feeders. Untreated colonies received 67% sucrose without fumagillin continuously throughout the same period.

## Data collection procedures

**Colony assessments.** Colonies were assessed at 11 time points between May 2014 and May 2016 (S5 Table). All assessments included a measurement of colony size, inspection for visible symptoms of disease, a search for the queen or indications of queen-rightness, and the collection of samples of adult bees from the broodnest.

**Colony population measurements.** Three methods of measuring colony size were employed [33–36] (S1 File, S1 Fig, S5–S8 Tables). For May, June, and August assessments in 2014, bee and brood populations were estimated visually. In May, June, and August 2015 and May 2016, a digital imaging method was employed. In November and April, due to the colder

weather, colony populations were measured less invasively by estimating the size of the cluster of bees (S1 File).

**Honey production and colony weights.**   Filled honey supers were weighed after removal from the colonies, and net honey contents were determined by subtracting either the empty weight of the super before placement on the colony (NAB) or an average empty weight (SAB). Colonies in PEI did not produce surplus honey that was harvested.

Feed stores before and after winter were determined by weighing colonies during the November and April inspections.

**Pollen collection.**   In 2014, pollen was collected once from each hive in Alberta during the canola bloom period. The traps were in place for a maximum of four days per colony. In 2015, in all three regions, a subset of 12 large colonies per apiary (three from each treatment group) was repeatedly sampled for 48-hour periods approximately every two weeks from mid-May until mid-September [37].

**Colony viability.**   Data and samples were collected as long as any bees remained. If no bees remained, or the colony was queenless and broodless at the end of the study (May 2016), the colony was considered to have become non-viable at the midpoint between the last verifiably queenright date (when either the queen or all stages of worker brood were seen), and the next inspection date. Data from dates when the colony was nonviable were excluded from analyses.

## Statistical analysis

Analyses were conducted using R version 4.12, and R Studio build 372 [38–43]. Additional details about the statistical models can be found in S1 File. The project data can be found in S2 and S3 Files, the R code in S4 File, and the output in S5 File.

Colony survival and queen survival were analyzed as Cox proportional hazards models using the R Survival package, version 3.3–1 [40, 42].

Measures of colony population (adult bee counts, sealed brood counts, and cluster sizes), productivity (honey production, pollen collection), and syrup feed consumption (colony weight) were analyzed as dependent variables in linear mixed effects models using the R nlme package, version 3.1_153.

Statistical significance (p) values for contrasts are shown without adjustment for multiple comparisons; instead, the significance threshold (p = 0.05) was adjusted with a Bonferroni correction. When the analysis of variance indicated that a factor and its interaction were both significant, we present contrasts for both. In such cases, the interactions were usually differences in the size of the treatment effect, not the direction, and as such the significant interaction does not exclude the more general result.

## Results

### Colony survival

More than half of the colonies (194 of 362; 53.6%) died during the study. Twenty-seven percent (32 of 117) of colonies in Southern Alberta, 47% (58 of 123) in Northern Alberta, and 85% (104 of 122) in PEI perished (Fig 1A; S9 Table). The high colony loss rate at PEI was partly associated with colony splitting and feeding practices employed by the cooperating beekeeper (S10 Table). Colonies that were split in summer 2014 were more likely to die during winter ($\chi^2$ = 6.70, df = 1, p-value = 0.010).

Protein supplemented colonies were more likely to die in all regions (ANOVA main effect: $\chi^2$ = 5.325; df = 1; p = 0.021) but on a within-region basis, this effect was significant only in Northern Alberta (interaction with region: $\chi^2$ = 6.493; df = 2; p = 0.039). Protein-

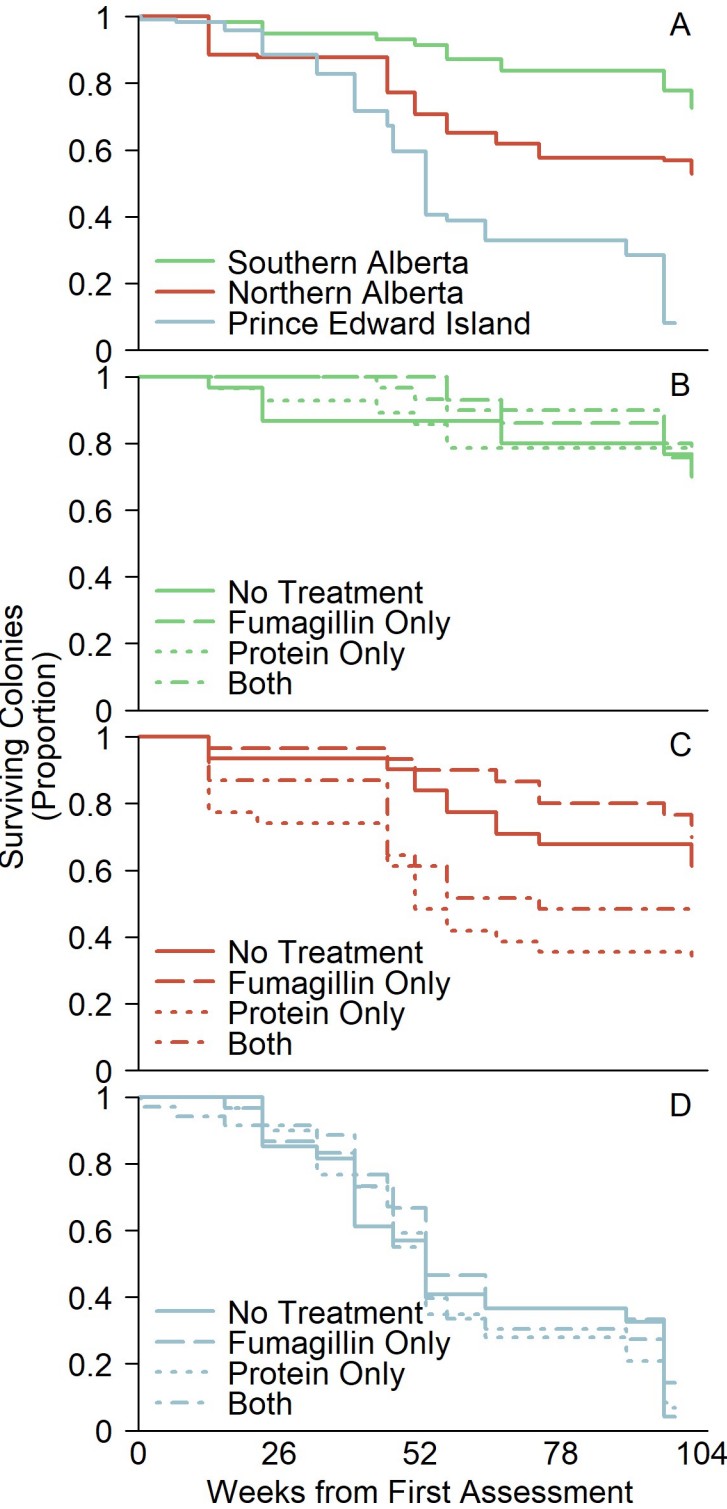

**Fig 1. Colony survival curves.** (A) Colony survival by region. (B) Effect of treatment on colony survival in Southern Alberta. (C) Effect of treatment on colony survival in Northern Alberta. (D) Effect of treatment on colony survival in PEI.

supplemented colonies had more than double the death rate in Northern Alberta (Fig 1C; hazard ratio: 2.45 ± 0.65; z = 3.36; p<0.001) but produced insignificant increases in the other regions (SAB: Fig 1B; hazard ratio: 1.02 ± 0.36; z = 0.058; p = 0.95; PEI: Fig 1D: hazard ratio: 1.14 ± 0.21; z = 0.72; p = 0.47). Fumagillin treatment was associated with a reduction in risk of survival, however this was not statistically significant ($\chi^2$ = 1.284; df = 1; p = 0.257; hazard ratio: 0.84 ± 0.13; z = -1.13; p = 0.26).

## Queen survival

Twenty percent of the original queens (63 of 316) survived until the end of the study. Thirty-eight percent of surviving colonies were still headed by their original queen (64 of 168; S9 Table); these numbers include splits made in 2014 but not those made in 2015. The greatest proportion of surviving original queens were in Southern Alberta (47; 73%). There were no differences in queen survival among treatment groups ($\chi^2$ = 1.87, df = 3, p = 0.60), however the pattern resembled that seen with colony survival. Relatively few of the surviving protein supplemented colonies retained their original queen (protein supplemented colonies: 33%; 24 of 73, unsupplemented colonies: 43%; 39 of 90). Colonies that received fumagillin without protein had the greatest queen survival (45%; 22 of 49).

The timing and number of queen events differed among regions (S9 Table). Some colonies experienced multiple natural queen replacements. A Cox Proportional Hazards test for queen survival, using only queen events that were not associated with colony death, did not reveal significant differences associated with the treatments, but the effect estimates resembled those for the Cox model of colony survival (hazard ratio for protein: 1.1; for fumagillin: 0.9; p>0.05 in both cases). Queens in protein supplemented colonies tended to have shorter survival times while those in fumagillin treated colonies had longer survival times. Because many colony deaths were known to have been preceded by a queen loss, we also considered a combined model in which all colony deaths were treated as a type of queen event (representing the case where queen replacement failed). In that model, protein supplements significantly reduced queen survival times regardless of region (ANOVA: $\chi^2$ = 6.619; df = 1; p = 0.010; effect estimate: hazard ratio = 1.26 ± 0.12; z = 2.52; p = 0.012). The effect of fumagillin still was not significant (ANOVA: $\chi^2$ = 1.463; df = 1; p = 0.227; effect estimate: hazard ratio = 0.87 ± 0.10; z = -1.21; p = 0.23).

## Colony population measurements

**Effects of region and date.** For each measure of colony population, differences related to region and date were highly statistically significant (p<0.001) and greatly outweighed the effects of treatments. For brevity, these comparisons are not reported in subsections below, but may be found in S5 File. *Varroa destructor* levels were well managed and not biased among treatment groups. Mites per hundred bees averaged over all dates were 0.02 (SAB), 0.08 (NAB), and 1.7 (PEI).

The colonies in Southern Alberta, which had been started as nucleus hives, were initially much larger than those in Northern Alberta, which had been started from packages (SAB: 9,200 ± 300 adult bees; NAB: 3,900 ± 130 adult bees; S2 Fig). Nevertheless, the Northern colonies had more brood (S3 Fig) and grew more quickly, reaching a peak of 25,000 ± 920 adult bees per colony in August 2014. Changes in colony populations during winter (cluster size; S4 Fig) corresponded to the length and timing of the season. Cluster measurements were not equivalent among regions because of differences in temperature and the timing of winter. After winter, colonies in Southern Alberta were by far the largest and colonies at PEI were the

smallest, while Northern Alberta colonies, which experienced the longest period of cold weather, also experienced the largest reduction in size.

Regional patterns of colony growth in the second year mirrored the first with the exception that that, overall, surviving colonies were larger (S2–S4 Figs).

**Effects of spliting in PEI.** The late summer splits at PEI created a complicating factor. Though the experimental protocol required nucleus colonies be created in spring 2014, the beekeeper also split colonies in late summer, in keeping with their standard management. The initial adult bee population in PEI was low (6,800 ± 290 bees per hive in June 2014), but these colonies had nearly as much sealed brood as the Northern Alberta colonies and the average colony population would have exceeded 14000 bees by mid August 2014 (calculated by ascribing all adult bees in the splits to their respective parent colonies). After splitting, the average colony in PEI contained only 9,700 ± 450 adult bees in August 2014. Daughter splits had only open brood at the time of inspection while the parent colonies had the most brood of any site or date (18,100 ± 770 sealed brood cells).

Because the splits were a beekeeper decision, the treatment groups were not equally affected. Protein supplemented colonies were split more frequently (a marginally non-significant effect when both years are combined: $\chi^2 = 3.2$, df = 1, p = 0.07). The decision to split colonies would presumably have been based on perceived colony size, but our measurements do not support the view that protein supplemented colonies were larger before splitting. For example, in August 2014, if all adult bees in the splits are attributed to the parent colony, adult populations in PEI before splitting were 14,300 ± 820 in the protein supplemented group and 13,900 ± 860 in the unsupplemented group, which is not a significant difference (two-sided t test: t = 0.384, df = 73.9, p-value = 0.70).

In both years, colonies that had been split were significantly smaller in November than colonies that had not been split (2014: t = 3.40, df = 70.2, p = 0.001; 2015: t = 2.06, df = 46.4, p = 0.045), and among split colonies, daughters were significantly smaller than parents in 2014 (2014: t = 5.62, df = 40.1, p<0.001; 2015: t = 1.37, df = 18.4, p = 0.18). Colonies that were split weighed less before winter than colonies that were not split, and daughter colonies from splits weighed less before winter than parent colonies (2014, not split versus split: t =. 5.23, df = 66.2, p<0.001; 2015, not split versus split: t = 1.71, df = 32.12, p = 0.097; 2014, daughter versus parent: t = 2.02, df = 58.0, p = 0.049; 2015, daughter versus parent: t = 1.71, df = 32.1, p = 0.31).

**Effects of treatments.** Protein supplements did not increase the adult bee count in any region at any measurement date (S2 Fig). On the contrary, protein supplements reduced the number of adult bees later in the experiment (interaction of protein supplements and date: F = 2.99; df = 5,1115; p = 0.011). Averaging the last five adult bee count dates across all three regions, supplemented colonies had 1,020 ± 410 fewer adult bees per colony (t = -2.48, df = 350, p = 0.014). This difference was not statistically significant on any individual date except August 2015 (-2,150 ± 650 adult bees; t = -3.31, df = 350, p = 0.001). By region (Fig 2A), the negative effect of protein supplements on adult bee population was significant only in Northern Alberta. Protein supplemented colonies in Northern Alberta had fewer adult bees across all dates after June 2014 (-1,760 ± 680 adult bees per colony; t = -2.584, df = 350, p = 0.010) and in August 2015 alone (-2,890 ± 850; t = -3.386, df = 350, p<0.001).

Early in the experiment, in June 2014, protein supplemented colonies produced more sealed brood than standard colonies across all regions combined (+1330 ± 400 sealed brood cells per colony; t = 3.34, df = 333, p < 0.001) and in Southern Alberta alone, this difference was substantial (difference by contrast: +1,960 ± 460 sealed brood cells per colony; t = 4.30, df = 333, p<0.001). On all other dates, however, protein supplemented colonies had, on average, less brood (protein * date: F = 5.37, df = 5,1091, p<0.001). Combining the last 5 brood inspection dates of the study, protein supplemented colonies had less brood in all regions

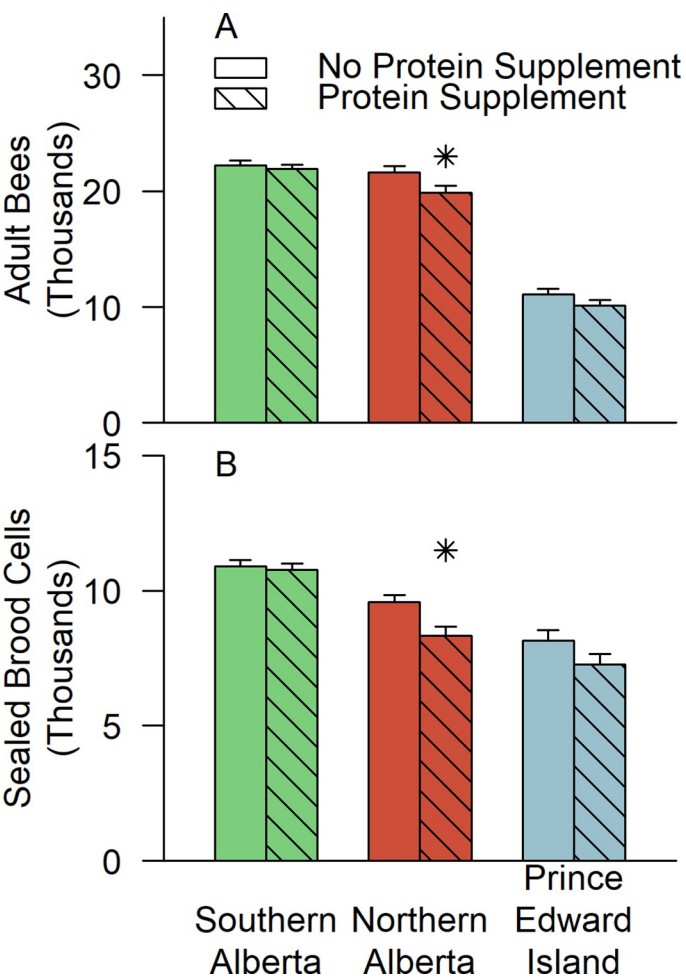

**Fig 2. The effect of protein supplements on colony size late in the experiment.** Data shown are estimated marginal means for (A) adult bee population and (B) sealed brood population averaged over the levels of the fumagillin treatment and the last five assessment dates (i.e., August 2014, May 2015, June 2015, August 2015, and May 2016). Stars (*) above a column indicate that the supplemented group was significantly different from the unsupplemented group within region (p<0.05, Bonferroni adjusted; see S5 File).

(-750 ± 250 sealed brood cells; t = -2.99; df = 333, p = 0.003) and particularly in Northern Alberta (-1,250 ± 410 sealed brood cells; t = -3.02, df = 333, p = 0.003; Fig 2B).

Fumagillin treatments were applied each fall between the August and November inspections. The three-way interaction of region, date, and fumagillin was significant in models for adult bees and sealed brood (Adult Bees: F = 1.96, df = 10,1115, p = 0.034; Sealed Brood: F = 2.01, df = 10,1091, p = 0.029). Significant effects were detected by contrasts only in June 2015, when fumagillin-treated colonies in Northern Alberta contained more bees and brood than untreated colonies, while treated colonies in PEI contained fewer bees and brood than untreated colonies. Neither effect reached significance in contrasts within region and date when averaged across levels of the protein supplement treatment, after accounting for multiple comparisons. However, in Northern Alberta, among colonies that were not receiving extra protein supplements, fumagillin was associated with significantly larger adult bee populations. The surprising negative effect of fumagillin on colony populations in PEI in June 2015 arose entirely among colonies which had been split in August 2014. There were more splits in the fumagillin treated group and the fumagillin-treated split colonies were smaller both before and

after winter. Splitting and fumagillin supplied in sugar syrup (see "Colony weight", below) both reduced the amount of feed stored by colonies prior to winter, which made these colonies particularly vulnerable, although death rates were similar. Two-thirds of fumagillin-treated splits weighed less than 40 kg prior to winter.

Cluster sizes were larger in the second winter than in the first. Fumagillin did not affect cluster sizes (F = 0.358, df = 1,235, p = 0.550). In November 2014, in Southern Alberta, protein supplemented colonies were 1.45 ± 0.40 inter-frame spaces larger (t = 3.649, df = 235, p < 0.001) than standard colonies. However, the four-way interaction of protein, region, year, and month was not significant in the analysis of variance.

In the first year, protein supplemented and standard colonies were the same size (averaged across levels of month, region, and fumagillin treatment; Fig 3A), but the supplemented colonies were 0.87 ± 0.33 inter-frame spaces of bees smaller in the second winter (t = -2.65, df = 235, p = 0.009). Additionally (Fig 3B), supplemented and standard colonies were similarly sized in fall (averaged across levels of year, region, and fumagillin treatment) but by spring, protein supplemented colonies were 0.66 ± 0.32 inter-frame spaces of bees smaller (t = -2.07, df = 235, p = 0.039). The change in cluster size during winter also approached significance. Colonies that had received protein supplements declined 0.66 ± 0.34 inter-frame spaces of bees more than standard colonies (t = 1.93, df = 235, p = 0.055).

A significant two-way interaction between region and protein supplements was also detected (F = 4.00, df = 2, 235, p = 0.020). No within-region contrast reached significance, but supplemented colonies were relatively large in Southern Alberta and smaller at the other two locations.

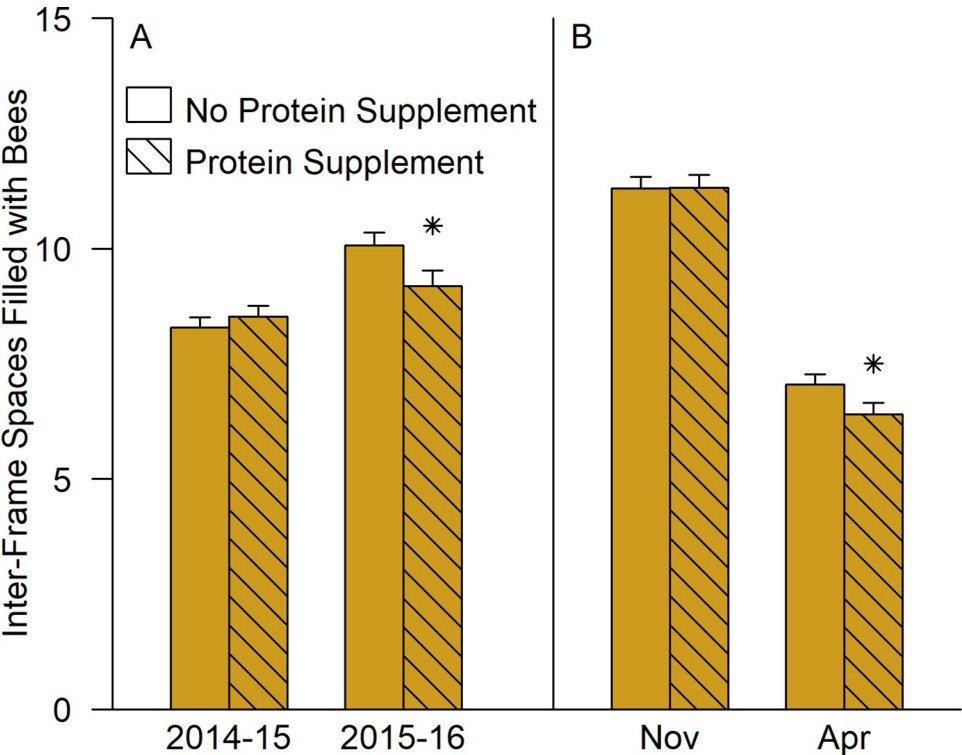

**Fig 3. The model effect of protein supplements on cluster size (mean ± SE).** Estimated marginal means are shown averaged across the levels of region and fumagillin. (A) Interaction of protein supplements and year (averaged across month): (B) Interaction of protein supplement and month (averaged across year). Stars above a supplemented column indicate a significant difference from the corresponding unsupplemented group (p<0.05, Bonferroni adjusted; see S5 File).

**Colony weight.**   Colony weights were measured at the November and April inspection dates as an indicator of feed storage and consumption. Colony weights were greater in fall, and in the second year of the study, and varied among regions according to the feeding practices of the beekeeper (S11 Table). Colonies in PEI weighed far less than either Alberta site in the first winter, and many were near starvation or had starved by spring 2015. In PEI, colonies that were split were lighter than colonies that were not split, and daughter colonies were lighter than parent colonies (S10 Table).

Protein supplemented colonies were 2.01 kg ± 0.57 kg (t = 3.513, df = 235, p<0.001) heavier than standard colonies before winter, but not after (Fig 4), indicating that more feed was stored and consumed. Fumagillin, in contrast, reduced average colony weight (S5 Fig) (F = 23.77, df = 2, 235, p < 0.001) but its effect size ranged from zero to 5 kg per colony depending on

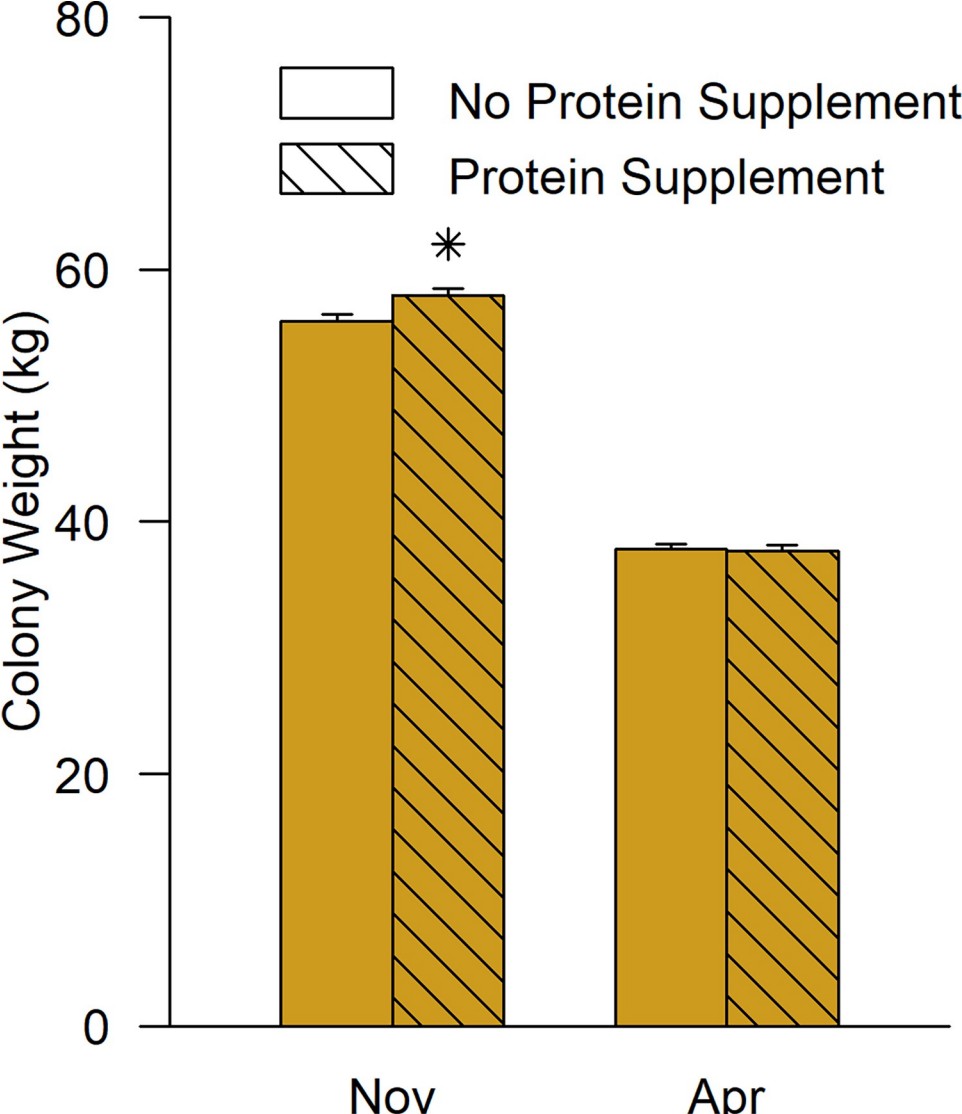

**Fig 4. The model effect of protein supplements on colony weight (mean ± SE).** Estimated marginal means are shown averaged across region, year, and levels of the fumagillin treatment. Stars above a column indicate that the supplemented group was significantly different from the unsupplemented group (p<0.05, Bonferroni adjusted; S5 File).

region (F = 5.735, df = 2, 235, p = 0.004) and there was a three-way interaction with year and region (F = 10.71, df = 2, 551, p < 0.001). The effect of fumagillin did not depend on month, which indicates that fumagillin reduced feed acceptance and storage, but not consumption during winter. One of the six within-region-and-date contrasts was statistically significant (Southern Alberta, first winter; t = 4.875, df = 235, p < 0.001).

**Honey production.** Colonies in Northern Alberta produced more honey (110 ± 3 kg per colony) than those in Southern Alberta (25 ± 1 kg per colony) in 2014 (t = 23.6, df = 228, p<0.001). Protein supplements did not affect honey production in 2014 (t = -0.593, df = 228, p = 0.55; fumagillin had not yet been applied), but in 2015 there was a significant treatment by site interaction. In Northern Alberta, colonies that received fumagillin but not excess protein supplements produced 23.5 ± 8.4 kg more honey per colony than colonies that received neither treatment (Fig 5; t = 2.82, df = 177, p = 0.005).

**Pollen collection.** Pollen collection preceded fumagillin treatment in 2014, and fumagillin had no effect on the weight of pollen collected in 2015; as such, fumagillin treatment was dropped from the model. Since protein supplements were not applied during the bloom period of canola, all pollen collection measurements in 2014 and mid-season measurements in Alberta in 2015 measure the effect of recent, but not current, protein supplementation. In 2014, protein-supplemented colonies in Southern Alberta collected 2.4 ± 1.1 g less pollen per day (S6 Fig; t = 2.246, df = 5, p = 0.075) than colonies that had not been supplemented. In Northern Alberta, protein supplemented colonies collected slightly more pollen, but the difference was not significant (t = 1.27, df = 4, p = 0.27).

In 2015, pollen was trapped repeatedly from a subset of colonies in each apiary. Overall, protein supplemented colonies collected less pollen during the early and middle parts of the season, but more pollen after August 10th (S7 Fig; two-way interaction of protein and season: F = 3.37, df = 2, 560, p = 0.035; no contrasts of protein treatment within season were significant).

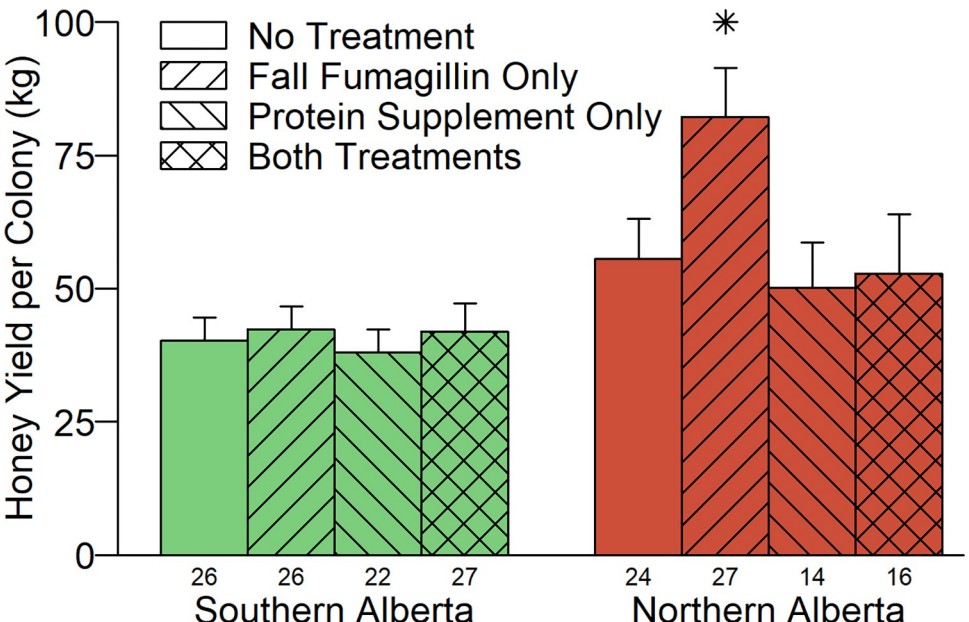

**Fig 5. Honey yield per colony in summer 2015, by region and treatment group (mean ± SE).** Stars above a column indicate that the treated group was significantly different from the untreated group (p<0.05, Bonferroni adjusted; see S5 File). The number of colonies that were still viable within each treatment group is shown beneath the columns. No surplus honey was produced in PEI.

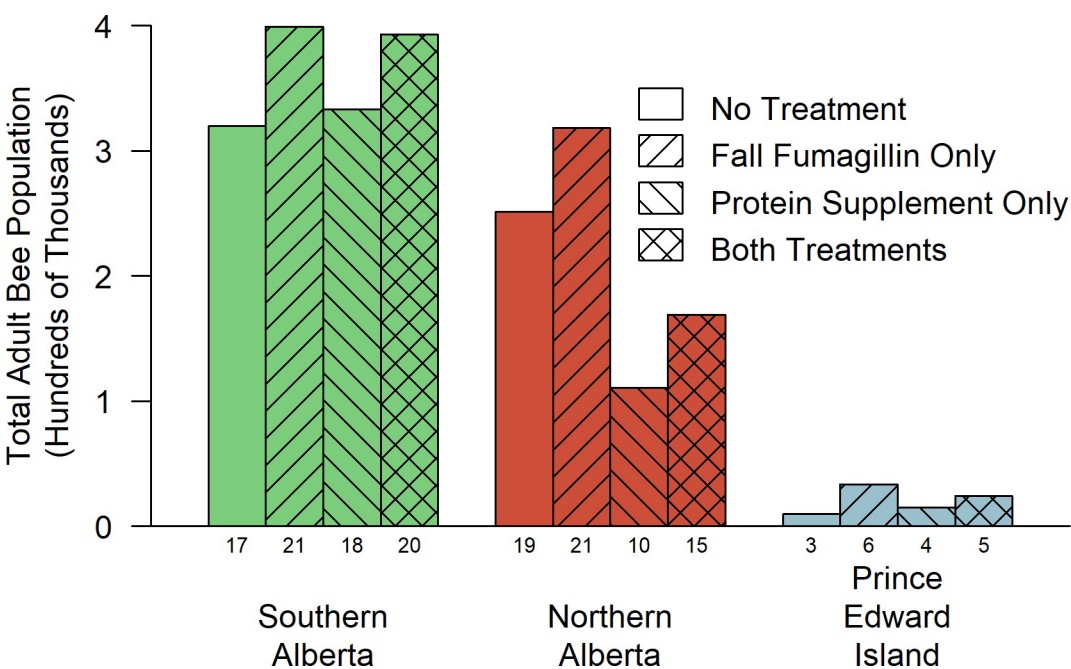

**Fig 6. Total adult bee population after two years of treatment.** The sums of the adult bee populations of all surviving colonies on the last inspection date of the experiment, shown by treatment group and region. Numbers beneath the columns indicate the number of viable colonies remaining.

**Colony size—an alternative view.**   Our analysis has treated colony survival, queen survival, and various measures of colony size and productivity as discrete outcomes, but in fact, they are likely to be related. In particular, factors that affect worker bee health, and therefore colony size and productivity, are likely also to affect the risk of queen or colony death. If a management strategy increases the size of viable colonies and also reduces the risk of colony death, either measure in isolation will underestimate the effectiveness of that strategy. Fig 6 provides a simple way to test whether this may have occurred, in which all the bees in each treatment group and region were summed on the last inspection date of the study. From this, three findings readily became apparent:

1. Local factors had a larger effect than either the fumagillin or protein supplement treatments on the population outcome.

2. Although we detected main effects related to protein supplements, the detrimental effects occurred primarily at the NAB site.

3. Although we detected only temporary and local effects from fumagillin, by the end of the study, every fumagillin-treated subgroup had considerably more bees than the corresponding group that was not fumagillin-treated.

## Discussion

In this paper we report the effects of two often-recommended beekeeper interventions on cohorts of honey bee colonies in three distinct climatic regions with economically different beekeeping operations. The first and clearest of our observations is that the effects of the interventions were small compared to the differences among the beekeeping operations, and were also small compared to the differences among colonies within an operation. We needed the

full statistical power of this large, repeated-measures study to detect them. For example, one of our major findings is that protein supplements unexpectedly reduced adult bee populations over time. Yet after two years of treatment, protein supplements had reduced the average colony size in Northern Alberta by only about 1,500 bees, which was less than 1/4 of the standard deviation among colonies there at that time (6,300 bees) and just over 1/10th of the average colony size (13,000 bees). It is apparent that location, general management, and differences among individual colonies at the same site are far more influential than the two treatment strategies we tested.

## Fumagillin

Fumagillin was first reported to be effective against *Vairimorpha apis* infections in honey bees in 1952 [44], and subsequently came to be recommended for widespread prophylactic use, especially in the commercial queen industry and in the hiving of package bees [27]. Subsequently, cage trials and field trials clearly showed that fumagillin kills both species of *Vairimorpha* that infect honey bees [23, 24, 32, 45]. Despite this finding, some reviewers have recently expressed skepticism about the effectiveness of fumagillin [28, 29, 46]. Others have challenged the importance of *V. ceranae* as a pathogen [26]. Some studies have reported no significant benefits following fumagillin treatment [30, 47, 48], while others have reported large effects [31, 49]. The suspicion that fumagillin, though it kills *Vairimorpha*, may not produce reliable economic benefits is a serious concern for beekeepers.

Contrary to historical recommendations [27], we did not find evidence that fumagillin significantly reduces the frequency of queen loss and the effect of fumagillin on colony survival, though positive, was also not significant. We identified a potential negative effect of fumagillin at the colony level. Colonies accepted less feed when fumagillin was in the syrup, thus increasing the risk of starvation. Although this effect was only significant in Southern Alberta in 2014, we suspect that our data underestimate the true effect of fumagillin on feed acceptance. Colonies were provided additional unmedicated feed after the fumagillin treatment, following the beekeepers' normal practice, and were not weighed until the end of the feeding period. Weights of fumagillin-treated colonies were lower in every region-date combination except for SAB in 2015, which was also the only case where the entire fumagillin dose was provided as a drench treatment rather than in feed.

Despite the above concern, overall, our results support the view that fumagillin has a positive effect on honey bee colony health. Fumagillin did not affect average brood or bee populations on most dates. However, in Northern Alberta, fumagillin was associated with significantly higher adult bee populations in June 2015 and subsequently with significantly higher honey production. In addition, by the end of the two-year study, fumagillin-treated groups in all regions had considerably more bees in total than the corresponding untreated groups.

## Protein supplements

Field studies have not consistently shown that commercial protein supplements are beneficial to honey bee colonies [20], although benefits have been detected in specific cases [18, 50]. Protein supplements support brood rearing in spring in years when pollen availability is a limiting factor, but not in all years [50], and many of the commercial supplements may be about equally effective [18] in promoting short term colony growth. However, few if any studies have tracked the same colonies for longer than one year, so the long-term effect of protein supplementation is largely unknown. Many have suggested that modern agricultural land use is a cause of malnutrition for bees [5, 51], and as such we expected that if a highly nutritious protein supplement were provided continuously, colony performance might improve.

As expected, protein supplements initially increased brood production, however, the increase was trivial in two regions and never resulted in a larger adult bee population. In one region (SAB) protein-fed colonies produced significantly more brood in June 2014 and were larger than unfed colonies in November 2014. Nevertheless, that was the last time a positive effect was observed. Across all regions, protein supplemented colonies declined more during winter than unsupplemented colonies. Over the long term, supplemented colonies were smaller than control colonies (NAB, PEI) or were not improved (SAB), and in Northern Alberta there was a substantial increase in the risk of death.

It is important to note that our protein supplement treatment was over and above the standard supplementation for the region, and our "unsupplemented" colonies received some protein supplementation in early spring, as is standard beekeeping practice. With that caveat, we draw several conclusions from these observations. First, nothing in our data supports the hypothesis that inadequate pollen (either quantity or quality) is a major cause of honey bee colony losses or under-performance in these regions. Undoubtedly, poor nutrition *could* weaken or kill hives, but in regions with a successful bee industry the pollen supply and existing feeding practices are probably adequate, and not the cause of recent poor performance. Secondly, protein supplements increase the rate of brood rearing at certain times and places—notably Southern Alberta in the first year. Thirdly, protein supplements consistently led to lower adult bee populations. This suggests that the increased brood rearing may have been offset by a decrease in the average adult lifespan, and the harmful effect on the adults was more significant over the long term than the benefit to the brood.

We are not the first to suggest that protein supplements lead to shorter lifespans for adult bees. La Montagne et al. [52] fed two protein supplements to colonies in Quebec and reported no improvement in colony performance as a result. They examined the effect on adult bee lifespan using marked bees in the hive and found that both supplements reduced average adult lifespans. Cage trials have shown that when *Vairimorpha*-infected bees consume protein, two effects occur: (1) infected bees that receive protein supplements live longer than infected bees that did not receive protein supplements and (2) *Vairimorpha* spores replicate faster and to a higher maximum level in the midguts of these bees [53, 54]. It has been unclear how these results relate to colony level performance, but the following interpretation would be consistent with our observations: when an infected bee consumes protein, both the bee and the parasite benefit, however the benefit to the bee is limited to the life of that bee, while the additional parasites remain in the hive to increase the prevalence of the disease. During this study, we collected over 6000 samples of bees for analysis of pests and parasites, and from analyses of these samples, we intend to address this question in a future report.

The value of high quality, nutritious natural pollen for bee hives has been established beyond any possible doubt [55]. In this study, the patties contained 25% pollen, which is far more than most commercial formulations, but they did not produce the expected benefits. Four possible explanations occur to us: (1) as mentioned, the colonies may not have been pollen-limited; (2) there might be some anti-nutritional compound or pesticide, unknown to us, remaining in the formulation; (3) periods of reduced pollen consumption (and hence reduced brood rearing) might be required for natural disease and pest resistance in the honey bee; or (4) the unnatural way in which protein supplements are supplied to the hive may have unintended consequences. Pollen that is naturally collected by the bees is stored as bee bread and is consumed by both nurse bees and larvae [56]. Noordyke et al. [57] recently examined the fate of protein supplements fed as patties in the hive, and found that the supplements were not stored as bee bread or fed directly to larvae; they were apparently consumed almost entirely by the adult bees.

## Regional differences

Hives belonging to the Southern Alberta seed canola pollinator had by far the greatest survival (73% survived two years) and the most stable colony populations. The hives grew slowly during summer but declined only slightly in winter. In the first year these bees benefited from the extra protein supplement, which suggests their environment was slightly pollen-limited in spring and possibly fall 2014, but since the difference was small and transient, it was unlikely to have justified the cost of treatment.

Hives from the Northern Alberta honey producer, which was the bee research laboratory of AAFC's Beaverlodge Research Farm, produced far more honey and pollen than those from SAB, especially in the first year, as is typical of colonies in this region [58]. Colonies in NAB grew much faster each summer, but also declined much more during winter. The quantity of trapped pollen dropped to near zero after the first week of August, and remained low; brood rearing declined more slowly, but was predominantly complete by the end of September. Despite the near-complete absence of incoming pollen in fall, these hives did not benefit from the protein supplement, which suggests that they were not protein limited.

The poorest performing hives in the study were those of the PEI blueberry pollinator, nearly all of which died over the two-year study. The poor performance of these hives, in part, may have been due to the study protocol. We required nucleus colonies to be started as early in the year as possible in order to match the treatments used in Alberta. However, the cooperating producer had a regular practice of splitting colonies in August, after blueberry pollination, and carried out that practice on the experimental colonies which had already been split. Thus, even though these colonies had the highest rate of brood production observed in the study, they were extremely small in fall and the splits had a higher-than-average rate of death. Nevertheless, the study protocol was likely only an incidental contributor to colony mortality in the region. Also noteworthy was the larger number and variety of visible disease symptoms in these colonies than anywhere else, higher varroa mite infestations, and during the first year of the study, colonies that were inadequately fed prior to winter resulting in high rates of starvation.

We did not find evidence that colonies of the blueberry pollinator suffered from inadequate forage. The very high rate of brood production of both supplemented and non-supplemented colonies in PEI, particularly in 2014, indicates that forage was not limiting in the region. In addition, if pollen sources during blueberry pollination had been inadequate, protein-supplements should have produced an increase in sealed brood cell count in June 2015. No such effect was found. Nor did protein supplements affect the amount of pollen collected by colonies in blueberries (S7 Fig, PEI before July 1).

## Conclusions

We have examined the long-term effects of two beekeeper interventions on honey bee colony health. Protein supplementation and fumagillin treatment both produced benefits in specific situations, but not in general. Local and individual differences among colonies were far larger than the effects of either treatment. There appears to be no risk associated with ordinary fumagillin treatment, provided the hive has adequate feed, and in some circumstances, there are substantial benefits. It seems reasonable to expect that the benefits of fumagillin are greatest when infections with *Vairimorpha* spores are severe, which may be more likely in with regions with a long, cold, temperate winter. Beekeepers should be cautioned against applying protein supplements when there is not a clear protein shortage. It may not be economically beneficial and under certain circumstances, the supplement may lead to unintended harm.

## Supporting information

**S1 File. Methods details.**
(ODT)

**S2 File. Main dataset.**
(CSV)

**S3 File. Pollen 2015 dataset.**
(CSV)

**S4 File. Statistical analysis code.** R markdown file with the code used to produce the statistical output file. Requires RStudio, R (necessary R packages are listed near the top of the file), and both dataset files. To run, the S2–S4 Files should all be saved to the same project folder.
(RMD)

**S5 File. Statistical analysis output file.** HTML file showing the output of the statistical analysis.
(HTML)

**S1 Fig. Colony population measurements.**
(POT)

**S2 Fig. Colony size measured as adult bees per colony (mean ± SE).** Stars (*) above a column indicate that the treatment group was significantly different from the untreated control group in contrasts within region and date ($p < 0.05$, Bonferroni adjusted; see S5 File). Numbers below a column indicate the number of viable colonies in the treatment group. Fumagillin was first applied in the fall of 2014; consequently, only two columns (untreated and protein supplemented) are shown on the earlier dates; and these columns include colonies subsequently treated with fumagillin.
(TIFF)

**S3 Fig. Colony size measured as sealed brood cells per colony (mean ± SE).** Stars (*) above a column indicate that the treatment group was significantly different from the untreated control group in contrasts within region and date ($p < 0.05$, Bonferroni adjusted; see S5 File). Numbers below a column indicate the number of viable colonies in the treatment group. Fumagillin was first applied in the fall of 2014; consequently, only two columns (untreated and protein supplemented) are shown on the earlier dates; and these columns include colonies subsequently treated with fumagillin. Data shown for PEI in August 2014 reflect parent colonies only; daughter colonies had no sealed brood.
(TIFF)

**S4 Fig. Colony size measured as the number of inter-frame spaces filled with clustering bees (mean ± SE).** Stars (*) above a column indicate that the protein supplemented group was significantly different from the unsupplemented group in contrasts within region and date ($p < 0.05$, Bonferroni adjusted; see S5 File). Numbers below a column indicate the number of viable colonies in the treatment group. Fumagillin did not affect cluster sizes; therefore, columns include both fumagillin treated and untreated colonies.
(TIFF)

**S5 Fig. The model effect of fumagillin on colony weight (mean ± SE).** Estimated marginal means are averaged across levels of protein treatment and month because there was no significant interaction between fumagillin and these factors. Each combination of region and year is shown separately because there was a significant three-way interaction of fumagillin, region, and year. Stars (*) above a column indicate that the fumagillin-treated group was statistically different from the corresponding untreated group ($p < 0.05$, Bonferroni adjusted; see S5 File).

SAB:Southern Alberta; NAB: Northern Alberta; PEI: Prince Edward Island.
(TIFF)

**S6 Fig. The effect of protein supplements on the quantity of trapped pollen per day in 2014 (mean ± SE).** Pollen was trapped for periods of two to four days at intervals during the canola bloom; that is, between the first week of July and early August.
(TIFF)

**S7 Fig. The effect of protein supplements on the quantity of trapped pollen per day in 2015 (mean ± SE).** Pollen was trapped from a subset of the largest colonies in each apiary and treatment group, at two-week intervals between mid May and mid September. SAB: Southern Alberta; NAB: Northern Alberta; PEI: Prince Edward Island.
(TIFF)

**S1 Table. Apiary-level groups of honey bee colonies, their locations, and dates of movement.**
(ODT)

**S2 Table. Replicates by location and treatment.**
(ODT)

**S3 Table. The protein supplement treatment: quantities and dates of supplement application.**
(ODT)

**S4 Table. Fall fumagillin treatments by region.**
(ODT)

**S5 Table. Colony assessment dates.**
(ODT)

**S6 Table. Dataset used to estimate the relationship between area measurements and counts of adult bees.**
(ODT)

**S7 Table. Dataset used to estimate the relationship between area measurements and counts of sealed worker brood cells.**
(ODT)

**S8 Table. Details of photographic image analysis of adult bees using the Honeybee Complete® 4.2 software.**
(ODT)

**S9 Table. Colony and queen survival to the end of the experiment.**
(ODT)

**S10 Table. Characteristics of split and non-split colonies at PEI.**
(ODT)

**S11 Table. Weights of honey bee colonies before and after winter.**
(ODT)

## Acknowledgments

We thank the participation of Landen Stronks and Gabriel Calixte, of Kiwi Brian's Honey, as well as Peter Dillon and Jasper Wyman and Son Canada, Inc. We also acknowledge Sean

Murray for his assistance in the conduct of the experiment in PEI. Thanks also goes to technical assistance from: Elena Battle, Jamie Malbeuf, Marika Viens, Zachary Wagman, Benjamin King, Breanalee Beer, Ian Johnson, Danielle Ediger, Justin Mufford, and Christopher Isaac. In addition, we thank both anonymous reviewers for their guidance in improving the manuscript.

## Author Contributions

**Conceptualization:** Stephen F. Pernal.

**Data curation:** Michael Peirson, Lynae P. Ovinge.

**Formal analysis:** Michael Peirson.

**Funding acquisition:** Stephen F. Pernal.

**Investigation:** Michael Peirson, Abdullah Ibrahim, Lynae P. Ovinge, Shelley E. Hoover, M. Marta Guarna, Andony Melathopoulos, Stephen F. Pernal.

**Methodology:** Shelley E. Hoover, Andony Melathopoulos, Stephen F. Pernal.

**Project administration:** Stephen F. Pernal.

**Resources:** Stephen F. Pernal.

**Supervision:** Stephen F. Pernal.

**Validation:** Michael Peirson, Shelley E. Hoover, Stephen F. Pernal.

**Visualization:** Michael Peirson, Stephen F. Pernal.

**Writing – original draft:** Michael Peirson, Stephen F. Pernal.

**Writing – review & editing:** Michael Peirson, Lynae P. Ovinge, Shelley E. Hoover, M. Marta Guarna, Andony Melathopoulos, Stephen F. Pernal.

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
