## [Decision Letter · Decision Letter 0]

23 Aug 2023

PONE-D-23-21321The effects of protein supplementation, fumagillin treatment, and colony management on the productivity and long-term survival of honey bee (Apis mellifera) colonies in CanadaPLOS ONE

Dear Dr. Peirson,

Thank you for submitting your manuscript to PLOS ONE. After careful consideration, we feel that it has merit but does not fully meet PLOS ONE’s publication criteria as it currently stands. Therefore, we invite you to submit a revised version of the manuscript that addresses the points raised during the review process.

We look forward to receiving your revised manuscript.

Kind regards,

Olav Rueppell

Academic Editor

PLOS ONE

Journal Requirements:

3. We are unable to open your Supporting Information file "Supplement 4 Statistical Analysis.Rmd". Please kindly revise as necessary and re-upload.

Additional Editor Comments:

The complex study design is more a weakness than a strength, despite the large sample size. I also agree with the reviewers that the comparison among different regions is difficult (due to many co-occurring differences that cannot be disentangled) and therefore should be minimized, in contrast to evaluation of supplemental feeding and Nosema treatments, which are valuable. Furthermore, I agree with both reviewers that the manuscript needs to become more accessible to readers by streamlining and shortening its presentation considerably, including the modeling. For methods and results, this means reorganizing and transferring some material to supplemental files. For the discussion, it means focusing on the meaningful messages and issues, as suggested by both reviewers. 

Reviewers' comments:

Reviewer's Responses to Questions

**Comments to the Author**

1. Is the manuscript technically sound, and do the data support the conclusions?

Reviewer #1: Partly

Reviewer #2: Yes

2. Has the statistical analysis been performed appropriately and rigorously? 

Reviewer #1: Yes

Reviewer #2: Yes

3. Have the authors made all data underlying the findings in their manuscript fully available?

Reviewer #1: Yes

Reviewer #2: Yes

4. Is the manuscript presented in an intelligible fashion and written in standard English?

Reviewer #1: Yes

Reviewer #2: Yes

5. Review Comments to the Author

Reviewer #1: General comments:

This study aimed at assessing the effect of supplemental protein feeding and antibiotic treatment against nosema disease on the survival, development and productivity of honey bee colonies. The study was conducted over a two-year period in three different regions of Canada. The results were inconsistent across regions and seasons. In some regions treatments had a positive effect whereas in others, the same treatments (particularly supplemental feeding) had a negative effect on the variables measured. These results are expected because there were many uncontrolled variables that affected the performance of the colonies, like differences in geographic location, climate, colony management, time of year when the colonies were established, colonies that were divided once versus twice a year, etc. The value of the study is that it used many colonies (>350) and that it was conducted for two years (usually these type of studies are conducted for only one year). However, the authors should consider toning down their conclusions and warning the reader that these results (particularly those related to the negative effects of protein supplementation) should be taken cautiously and that future studies should be planned to try to standardize experimental conditions in a way that experimental error can be minimized.

Specific comments:

1. For future submissions please number lines and pages for easier review.

2. Introduction, page 11. Please italicize Varimorpha (scientific name).

3. Methods, page 13. It is not clear from the text if the colonies of all three regions used the same queen source (Kona queens). Please clarify.

4. Methods, page 14. It is not clear how PEI beekeepers supplemented their colonies. Please elaborate on the matter.

5. Results and discussion. Winter colony mortality was higher in Northern Alberta (54%) and colonies were less populated. The authors attribute those results to the effect of protein supplementation. However, it is not well explained why supplemental feeding would be detrimental to populations and colony survival. It is important to hypothesize as to why this could be. Additionally, please discuss that rather than the effect of the supplemental treatment, these detrimental effects could have been caused by V. destructor infestations. There is plenty of evidence that mite infestations are more frequently associated with colony losses during winter than any other cause. If possible, show data of mite levels for all regions and seasons, which could shed light on the results and conclusions of this study.

6. Page 40. Consider using another word or deleting the word “dramatically” related to honey production. Not scientific jargon. Was it significant? If so, please use that word.

7. Page 45. Colonies accepted less feed with fumagillin contrary to what occurs in cage experiments. Why is that? In cage experiments bees fed fumagillin tend to consume more syrup. Please try to explain this discrepancy.

8. Page 47. “Poor nutrition kill hives.” Hives are made from wood or other materials and are not live organisms. Do you mean colonies? If so, please reword.

9. A very high proportion of the original queens were lost or superseded. The period of queen replacement has a direct impact on colony populations. Does queen supersedure had to do with results on colony populations? Please discuss.

10. Tables. Please reduce the font size of subscript legends.

11. Table 3. The weight difference results from differences in feed and bee populations, not only feed weight. Please mention that.

12. Figures. Please use different colours for bars representing different treatments. The same colours are confusing.

13. Authors should consider limiting the paper to show results of the region or regions that had experimental colonies with more uniform and consistent management. For example, only Northern or Southern Alberta or both. That would reduce the size of the paper and will also provide data from colonies that were less affected by the uncontrolled variables. The conclusions would be more reliable in my opinion.

14. Please shorten the paper by at least one third. It is unusual to have manuscripts of more than 45 pages. Each section of the paper could be shortened. There are too many figures too (12). Consider presenting only the most representative data in figures and tables. If necessary, present some of the data in supplemental tables and/or figures to reduce the number of tables and figures in the manuscript.

Reviewer #2: This study examines the effects of two treatments (fall Fumagillin B treatments against Variamorpha spp. (Nosema spp.) and continuous pollen patty supplements) on long term productivity and colony survival in Canadian commercial colonies. Both treatments are applied prophylactically to reduce disease incidence and improve colony productivity and survival in colonies in many regions. The authors found that fumagillin treatments and pollen patty treatments have little to negative long term effects on colony productivity and survival, despite their reputation to the contrary among beekeepers. The study follows and evaluates these colonies very thoroughly and extend well beyond the period usually covered by similar studies. These findings are important to relate because they demonstrate neutral to negative effects of perhaps the two most recommended colony treatments aside from Varroa mite treatments. Despite major regional and management differences, an absence of benefits is observed for both treatments across all three regions. Studies such as this that report neutral to negative effects of common treatments are essential to public discussion of their value and should be reported.

The main problems of the study stem from exactly what the authors attempt to capture, namely, treatment effects with regional differences in management practices. Not surprisingly, the authors report that regional differences swamp treatment effects of fumagillin and pollen protein supplements. One real problem is that the different regional practices regarding colony establishment (packages, splits, nucs) start the colonies with very different sized populations and compositions, which in a sense make head on comparisons with performance data somewhat tenuous (apples to oranges comparisons). Unfortunately the observed regional differences could be attributed to different colony establishment practices as well as environmental differences. The study is particularly confounded by the late summer splits performed by the PEI beekeeper on what seem to be colony strength based criteria. This beekeeper action is understandable to anyone who has worked on longitudinal studies with commercial beekeepers (sometimes some of them do things that work against the experimental design). However, the claims that colony size could be predicted by adding original and daughter split colonies together are somewhat of a stretch. Problems notwithstanding, what is striking though is that a lack of positive benefits were observed in three quite different colony environments.

The manuscript itself is very extensive in presentation of modeling and modeling immediate results but light on discussion of these results to relevant literature. The modeling results are extensive framed and discussed in detail with thoughtful explanations of effect causes, but need to be streamlined to avoid overwhelming the paper. The number of figures (12) are quite high and some of them could be converted into supplemental figures to streamline the paper. The authors discuss their findings nicely with the relevant literature but may want to expand comparisons based on timing and duration of treatments (especially pollen patty substitution). The overall message that treatment efficacy varies regionally with environmental and regional practices is an eminently sensible one and should be the basis of whether such practices are locally useful for their intended purposes.

Specific comments

(general) The methods, results, and explanation of statistical approaches are very detailed and professional here. The authors handled a considerable amount of statistical comparisons with nuance and appropriate explanation.

(general, p47 1st sentence on protein supplement being above and beyond normal, continuous). The continuous protein supplementation is a critical aspect of the study that differentiates it from other pollen substitution studies. The authors emphasize this in the Abstract and in the Discussion, but this should be also be noted in the experiment outline at the end of the Introduction. Likewise, comparisons with other studies need to take into account the treatment timing and duration of those studies. The authors did a fairly decent job of this but it is worth going through the drive the distinction home.

(general) The authors need to be careful about presentation of non-significant trends and tendencies. Several times in the manuscript differences are presented before non-significant test results are presented. One suggestion would be to note quickly the absence of a significant difference but then discuss trends. I should note that this is simply a difference of presentation style and that the authors have been thorough and forthright in presentation of their results.

(general) Consider using the term “surviving colonies” to describe statistics gathered from surviving colonies only through the manuscript. Removal of data from colonies that later perished in the study is understandable but may lead to different results that if this data was left in at time points when these colonies were alive.

(general, p49-50) Blueberries have a mixed reputation in their impacts on pollinating honey bee colonies. In some regions, impacts appear on honey bee colonies appear to be severe. Any comments on colony survival in blueberry pollinating sites here, i.e. do the colonies often have trouble or is there sufficient forage to make it through?

p20 pollen collection p42 Fig 10. The methods and heading describes how pollen was trapped for canola pollinating colonies in 2014. Can you add details for the PEI colonies?

P23 Cluster size and colony weight models 2nd sentence The approach of dropping colonies that died during overwintering from the fall analysis is conservative but by choice exclude colonies in decline. How do the November comparisons compare if these doomed colonies are included?

p23 Results Colony survival 3rd sentence “104 of 122” The count of PEI colony failures appears to be off.

p27 Table 1 d count of queen replacements Not clear here on whether multiple replacements in a colony would be counted once or separately in these tallies.

p27 “…occurred during the winters (…, than during the summers …” Can you consider use of other terms to describe these halves of the year? Cold season and warm season come to mind (but are probably too awkward). Winters and summers seems to leave out fall and spring builds.

p28 Queen survival 4th sentence. Perhaps substitute “A lower proportion of colonies in the protein supplemented groups" for “Fewer colonies” given the disparity in colony survival.

Fig2 and Fig3 headings. For the PEI colonies, are the adult and brood population counts only from the original split colonies or both daughter and original colonies given the differences in bee splits here?

p33 The splits were a beekeeper decision … p34 Thus, it is probable that starvation … This is a realistic problem in longitudinal studies and well presented here.

p36 and p38 PEI colony weights and starvation Table 3. The high colony masses of the PEI colonies are quite interesting in light of the likely starvation of some of these colonies. The 2015-2016 colonies appear to be of near average mass of the SAB and NAB colonies. Did the PEI beekeepers attempt any supplemental sugar feeding as part of their overwintering management?

p44 Discussion 1st paragraph. The discussion of how regional differences were much stronger than treatment differences is illustrated quite well here. It is an honest point.

6. PLOS authors have the option to publish the peer review history of their article (what does this mean?). If published, this will include your full peer review and any attached files.

Reviewer #1: No

Reviewer #2: No

---

## [Author Response · Author response to Decision Letter 0]

2 Oct 2023

See uploaded file entitled "Response to Reviewers", as instructed in decision letter.

---

## [Decision Letter · Decision Letter 1]

20 Nov 2023

PONE-D-23-21321R1

The effects of protein supplementation, fumagillin treatment, and colony management on the productivity and long-term survival of honey bee (Apis mellifera) colonies

PLOS ONE

Dear Dr. Peirson,

Thank you for submitting your manuscript to PLOS ONE. After careful re-evaluation by the reviewers a few minor issues remain that I would like to encourage you to address. We invite you to submit a revised version of the manuscript that addresses the points raised during the review process.

We look forward to receiving your revised manuscript.

Kind regards,

Olav Rueppell

Academic Editor

PLOS ONE

Journal Requirements:

Reviewers' comments:

Reviewer's Responses to Questions

**Comments to the Author**

1. If the authors have adequately addressed your comments raised in a previous round of review and you feel that this manuscript is now acceptable for publication, you may indicate that here to bypass the “Comments to the Author” section, enter your conflict of interest statement in the “Confidential to Editor” section, and submit your "Accept" recommendation.

Reviewer #1: All comments have been addressed

Reviewer #2: (No Response)

2. Is the manuscript technically sound, and do the data support the conclusions?

Reviewer #1: Yes

Reviewer #2: Yes

3. Has the statistical analysis been performed appropriately and rigorously? 

Reviewer #1: Yes

Reviewer #2: Yes

4. Have the authors made all data underlying the findings in their manuscript fully available?

Reviewer #1: Yes

Reviewer #2: Yes

5. Is the manuscript presented in an intelligible fashion and written in standard English?

Reviewer #1: Yes

Reviewer #2: Yes

6. Review Comments to the Author

Reviewer #1: Thanks for addressing my concerns. The paper is much better now. Still a little bit long but I appreciate that the authors had made an effort to shorten this complex report

Reviewer #2: The authors have substantially revised their manuscript so that it is more compact, concise, and structurally easy to interpret. As noted by the Editor and reviewers, the main featured aspects of this study (regional comparisons, differences in protocols and local practices) were also highly problematic. At worst, the “telescoping” of effects from regional differences, beekeeper practices, and protocol differences could lead to overstatement of treatment effects. More so than studies of less substantial design, the manuscript needed to be trimmed, focused, and tightly structured in its presentation.

The revised manuscript achieves this balance. The authors refocused on treatment differences but also critically acknowledged key regional and beekeeper practices differences. The authors compartmentalized the presentation of the results in such a way that readers would not become overwhelmed or lose track of results. The refocus on treatment effects in the context of regional differences adequately wrangles an information dense presentation into its component sections. The number of figures, tables, and supporting information has been reduced both in the manuscript and by shunting large parts to the Supplemental Information sections. At the same time, the authors show openness and nuance in consideration of their design and results. They do not shy away from problems or flaws that arose during the experiment. The emphasis on reduced fumagillin treatment syrup intake as a possible stress factor (and it’s occurrence primarily in one region) is an area for management improvement.

I would suggest that the authors revisit their Figure and Supplemental Information citations carefully. Fig 1 and S1 Fig are missing from the reviewer’s copy of the manuscript but the corresponding results appear to be in line with data given in the Supplemental Information figures and files.

line 248 Are there stats to go with the Cox hazards on surviving queens?

line 264 The authors may want to slightly rephrase the section on mite infestation rates to support later statements on PEI colony disease/parasite stressors (paragraph starting on 618, specifically 629). This reviewer agrees that the average mite counts for PEI colonies (1.7 mites/100 bees) are marginally high (approaching treatment thresholds, which of course vary by region, local practices and time of year) and may be a cause for concern under the right conditions as noted in the Discussion.

line 273 Weren’t cluster sizes reported in S4 Fig not S3 Fig?

line 281 Should the supporting figures be S2 Fig, S3 Fig, and S4 Fig rather than S1 Fig, S2 Fig, and S3 Fig given the topic covered?

line 298 Marginally non-significant (p=0.07)?

line 560 An interesting feature of this study is that the pollen supplementation treatments occur in excess of local beekeeper practices (both forage and pollen supplementation) that presumably provide minimal nutrition in most years. One minor point – the pollen supply and pre-existing beekeeper supplementation practices are probably adequate to allow beekeepers to produce in these areas.

574 Authors may want to examine studies detailing increased Vairimorpha spore counts in hindgut in pollen fed bees for their companion paper. The idea that pollen feeding in supplements may temporarily directly benefit the feeding adult but not future generations is interesting, particularly if such supplementation leads to higher disease spore exposures.

7. PLOS authors have the option to publish the peer review history of their article (what does this mean?). If published, this will include your full peer review and any attached files.

Reviewer #1: No

Reviewer #2: No

---

## [Author Response · Author response to Decision Letter 1]

22 Nov 2023

See uploaded document entitled "response to reviewers".

---

## [Editor Report · Decision Letter 2]

4 Dec 2023

The effects of protein supplementation, fumagillin treatment, and colony management on the productivity and long-term survival of honey bee (Apis mellifera) colonies

PONE-D-23-21321R2

Dear Dr. Peirson,

We’re pleased to inform you that your manuscript has been judged scientifically suitable for publication and will be formally accepted for publication once it meets all outstanding technical requirements.

Kind regards,

Olav Rueppell

Academic Editor

PLOS ONE
---

## [Editor Report · Acceptance letter]

11 Dec 2023

PONE-D-23-21321R2 

The effects of protein supplementation, fumagillin treatment, and colony management on the productivity and long-term survival of honey bee (*Apis mellifera*) colonies 

Dear Dr. Peirson:

I'm pleased to inform you that your manuscript has been deemed suitable for publication in PLOS ONE. Congratulations! Your manuscript is now with our production department. 

Kind regards, 

on behalf of

Dr. Olav Rueppell 

Academic Editor

PLOS ONE